# Interaction between Enrofloxacin and Three Essential Oils (Cinnamon Bark, Clove Bud and Lavender Flower)—A Study on Multidrug-Resistant *Escherichia coli* Strains Isolated from 1-Day-Old Broiler Chickens

**DOI:** 10.3390/ijms25105220

**Published:** 2024-05-10

**Authors:** Sławomir Zych, Michalina Adaszyńska-Skwirzyńska, Małgorzata Anna Szewczuk, Danuta Szczerbińska

**Affiliations:** 1Laboratory of Chromatography and Mass Spectrometry, Faculty of Biotechnology and Animal Husbandry, West Pomeranian University of Technology in Szczecin, Janickiego Str. 29, 71-270 Szczecin, Poland; 2Department of Monogastric Animal Sciences, Faculty of Biotechnology and Animal Husbandry, West Pomeranian University of Technology in Szczecin, Janickiego Str. 29, 71-270 Szczecin, Poland; michalina.adaszynska@zut.edu.pl (M.A.-S.); malgorzata.szewczuk@zut.edu.pl (M.A.S.); danuta.szczerbinska@zut.edu.pl (D.S.)

**Keywords:** APEC, checkerboard, enrofloxacin, *Escherichia coli*, essential oils, MIC, synergy

## Abstract

Avian pathogenic Escherichia coli (APEC) causes a variety of infections outside the intestine. The treatment of these infections is becoming increasingly difficult due to the emergence of multi-drug resistant (MDR) strains, which can also be a direct or indirect threat to humans as consumers of poultry products. Therefore, alternative antimicrobial agents are being sought, which could be essential oils, either administered individually or in interaction with antibiotics. Sixteen field isolates of *E. coli* (originating from 1-day-old broilers) and the ATCC 25922 reference strain were tested. Commercial cinnamon bark, clove bud, lavender flower essential oils (EOs) and enrofloxacin were selected to assess the sensitivity of the selected *E. coli* strains to antimicrobial agents. The checkerboard method was used to estimate the individual minimum inhibitory concentration (MIC) for each antimicrobial agent as well as to determine the interactions between the selected essential oil and enrofloxacin. In the case of enrofloxacin, ten isolates were resistant at MIC ≥ 2 μg/mL, three were classified as intermediate (0.5–1 μg/mL) and three as sensitive at ≤0.25 μg/mL. Regardless of the sensitivity to enrofloxacin, the MIC for cinnamon EO was 0.25% *v*/*v* and for clove EO was 0.125% *v*/*v*. All MDR strains had MIC values for lavender EO of 1% *v*/*v*, while drug-sensitive isolates had MIC of 0.5% *v*/*v*. Synergism between enrofloxacin and EO was noted more frequently in lavender EO (82.35%), followed by cinnamon EO (64.7%), than in clove EO (47.1%). The remaining cases exhibited additive effects. Owing to synergy, the isolates became susceptible to enrofloxacin at an MIC of ≤8 µg/mL. A time–kill study supports these observations. Cinnamon and clove EOs required for up to 1 h and lavender EO for up to 4 h to completely kill a multidrug-resistant strain as well as the ATCC 25922 reference strain of *E. coli*. Through synergistic or additive effects, blends with a lower than MIC concentration of enrofloxacin mixed with a lower EO content required 6 ± 2 h to achieve a similar effect.

## 1. Introduction

Poultry are constantly exposed to microorganisms. Several important factors contribute to the relatively high level of microbial contamination in poultry farms, the most important of which are environmental factors (e.g., high temperature, dustiness, excessive moisture), poor hygienic quality of water and feed and lack of proper bio-assurance [1].

*Escherichia coli*, a Gram-negative facultative anaerobic rod-shaped bacterium, is part of the natural bacterial flora of the gastrointestinal tract of humans and animals and is therefore considered an important indicator of faecal contamination of water and food [2,3]. However, poultry colibacteriosis can develop as a primary or secondary infection, alongside other viral or bacterial infections [4]. Generally, about 10–15% of *E. coli* in the gastrointestinal tract of birds belong to the Avian Pathogenic *Escherichia coli* (APEC) serotypes [5]. The most common form of colibacteriosis in chickens occurs between 3 and 10 weeks of age, with a variety of symptoms, such as navel and yolk sac inflammation, acute sepsis, respiratory and reproductive colibacteriosis, cellulitis, arthritis and osteoarthritis syndrome [6]. Moreover, *E. coli* can penetrate the eggshell and spread to chicks during hatching, mainly causing yolk sac inflammation and acute septic colibacteriosis, resulting in early high mortality [7].

The treatment or prevention of colibacteriosis is mainly based on antibiotic therapy and autovaccines, but numerous studies indicate that multidrug-resistant strains of *E. coli* are common [8] and autovaccines are less effective and have not been widely used to date, mainly because APEC strains are very heterogeneous [9]. Imported one-day-old chicks (especially from various hatcheries) may be a source of new serotypes/strains with unknown antibiotic resistance and can be a potential source of dissemination of resistant bacteria in poultry production. In addition, the restriction on the use of antibiotics introduced by the EU Parliament and Council Regulation No. 2019/6 [10], also known as the “new veterinary regulation”, has forced the search for new ways to improve the level of biosafety.

As an alternative, essential oils (EOs) can be used either alone or in combination with common antimicrobial agents. Essential oils consist of approximately 20–60 volatile components, which are secondary metabolites produced by aromatic plants [11]. These volatile compounds (generally of low molecular weight below 500 g/mol) belong to various chemical classes, including terpenes, aldehydes, alcohols, ethers, ketones, esters, amines, amides and phenols [12]. The mechanism of antibacterial and antifungal actions of most of these components is not well established. The most popular opinion is that the interaction of hydrophobic components with lipids present in the cell membrane of microorganisms results in cell death [13]. The association of antibiotics with essential oils against resistant bacteria may expand the antimicrobial spectrum to reduce the emergence of resistant variants and minimize the use of a single antibiotic [14].

To select essential oils that are effective against *E. coli*, especially multidrug-resistant strains, a checkerboard study was performed including three selected commercial essential oils from spices (cinnamon, clove) and flowers (lavender) and simultaneously taking into account the positive interaction with one of the most commonly used antimicrobial agents in poultry enrofloxacin.

## 2. Results

The MIC results for enrofloxacin were in agreement with the disk diffusion method from the official test reports for the selected isolates (Table 1 as well as Appendix A). According to the recommendations in the VET01S, 5th ed. [15], the MIC interpretative criteria for enrofloxacin and *E. coli* in poultry (expressed in µg/mL) indicate that the isolate is resistant at an MIC ≥ 2 μg/mL and sensitive at ≤0.25 μg/mL of enrofloxacin (0.5–1 μg/mL is classified as intermediate). The results of the analysis are presented in Table 1 and are summarised in a gradient from isolates that are most resistant to enrofloxacin to those that are susceptible.

In the case of essential oils, each has its own best and usually constant individual MIC. Cinnamon bark EO was always effective at a concentration of 0.25% *v*/*v* (corresponding to 2.56 mg/mL; density 1.025 g/mL at 25 °C). Clove bud EO was always effective at two times lower concentrations than cinnamon—0.125% *v*/*v* (corresponding to 1.31 mg/mL; at a density of 1.05 g/mL). It is worth emphasising that the activity of these two oils was independent of the level of resistance of *E*. *coli* to enrofloxacin. A certain division was observed for the lavender flower essential oil. All multidrug-resistant isolates (MDR-1 to MDR-10) always had MICs for lavender at 1% *v*/*v* (equivalent to 8.79 mg/mL; at a measured density of 0.879 g/mL), whereas isolates with resistance to single antibiotic groups (SDR-1 to SDR-3) and drug-sensitive isolates (SENS-1 to SENS-3 and ATCC 25922) had MICs of 0.5% *v*/*v* (4.4 mg/mL of lavender EO).

Among the 153 checkerboards performed (17 *E. coli* strains × 3 combinations × 3 replications), no antagonism or even neutral interaction was found. The results are presented in Table 1 (best example of checkerboards; *n* = 51). The vast majority (64.7%) showed synergy between EOs and enrofloxacin. The remaining 35.3% of cases had additive effects. Cinnamon and clove EOs interacted similarly with enrofloxacin, with synergism noted more frequently for cinnamon EO (64.7%) than for clove EO (47.1%). The most common FIC index was 0.5 (further referred to as ‘weak’ synergy). It is characterised by a “stair-step” pattern on the plate, where the effective amount of both antimicrobials was reduced four times (referred to as 1/4 MIC of enrofloxacin and 1/4 MIC of EO) in relation to their individual MIC. A more detailed description is given in Appendix A. In contrast, lavender EO was much more prone to interact with enrofloxacin, especially as a strong synergy (82.35% of cases; (FICi = 0.155–0.375 and rarely 0.5). An example of strong lavender/enrofloxacin synergy is included (with the description) in the section Appendix A. The exceptions were the MDR-3 and SENS-3 isolates, as well as the ATCC 25922 reference strain, for which only additive effects were always recorded regardless of the essential oil used.

Unfortunately, even with synergy between enrofloxacin and EO, high resistance to enrofloxacin (MIC > 16 µg/mL) resulted in a situation in which these isolates still remained at the level of insensitivity to this antimicrobial agent. Only at the MIC for a minor degree of enrofloxacin resistance (2–8 µg/mL) did such isolates become intermediate or susceptible to enrofloxacin. Due to synergy, strains intermediate to enrofloxacin (SDR-1 to SDR-3) may become susceptible, while the effective concentration of enrofloxacin can also be significantly reduced among the susceptible isolates (SENS-1 and SENS-2). However, as the sensitivity to enrofloxacin increased, the importance of this interaction decreased. This is well demonstrated by the identical results for the SENS-3 isolate and the reference strain ATCC 25922, where only half of the effective enrofloxacin concentration was observed (i.e., a reduction from 0.016 µg/mL to 0.008 µg/mL).

Among the serotypes, the highest number of APEC O78 isolates (56.25%) was identified. Other APEC isolates were also identified: two isolates of O1 (12.5%) and one isolate of O2 and O18 (6.25% each). However, a large number of isolates had unknown serotypes (18.75%).

A time–kill assay was used to study the activity of selected antimicrobial agents (cinnamon, clove, lavender EOs and enrofloxacin, alone and in combination) against two bacterial strains (MDR-9 and ATCC 25922) to determine the bactericidal or bacteriostatic activity of an agent over time. The MDR-9 strain of *E. coli* was chosen because of its common drug resistance observed in studies of 1-day-old chicks (own experience; see also Appendix A) and common results of interaction between all EOs under study and enrofloxacin: “weak” synergy (FICi = 0.5) for cinnamon and clove and “strong” synergy for lavender (FICi = 0.375). Figure 1 shows the time–kill results for the MDR-9 strain (expressed as mean viable CFU/mL over time) for each EO and enrofloxacin at the MIC level (continuous lines (1) to (4)) compared with the control (MHB and MHB with 5% ACN; continuous lines (13) and (14)) and for each synergistic combination (short-dashed lines (5) to (7)) compared with their components applied alone (long-dashed two-pointed lines (8) to (12)).

All samples were inoculated with a shared suspension of MDR-9 strain (1.68 × 10^6^ CFU/mL; 0 h time point). Cinnamon EO at 0.25% *v*/*v* and clove EO at 0.125% *v*/*v* completely inactivated the MDR-9 strain within one hour (after 30 min, MDR-9 was barely detectable at 1.2 × 10^3^ CFU/mL and 2.8 × 10^3^ CFU/mL, respectively). In contrast, 1% *v*/*v* lavender EO required four times as long (i.e., 4 h) to reach this state (viable cells were still detectable after 2 h; however, only in trace amounts—5.03 × 10^2^ CFU/mL). Enrofloxacin at an MIC of 2 µg/mL, after an initially strong reduction in viable cells of the MDR-9 strain to 3.2 × 10^5^ CFU/mL within 1 h, was unable to kill this strain for a long time (average 2.2 × 10^4^ CFU/mL were still detected after 12 h). However, after 24 h of incubation, the viable cells were no longer detectable.

Among the three EO × enrofloxacin combinations studied, the most effective was the combination of lavender EO (1/4 MIC) and enrofloxacin (1/8 MIC), which killed this enrofloxacin-resistant strain within 4 h. The other two combinations (1/4 MIC of cinnamon or 1/4 MIC of clove and 1/4 MIC of enrofloxacin) required 8 h. However, the efficacy of all combinations derived from the checkerboard method was confirmed, especially when compared to that of slow-acting enrofloxacin administered alone. It is also noteworthy that all the components included in the synergistic blends when administered alone (lines (8) to (12)) had no bactericidal activity. In addition, visible turbidity of the culture appeared at the end of the incubation period. Bacterial growth without any antimicrobial agents (control samples 13 and 14) reached more than 1.0 × 10^10^ CFU/mL at 24 h which was manifested by the high turbidity of the sample in the tube. The same graph, but on a full logarithmic scale, is available as Appendix A.

Similar but significantly faster effects of antimicrobial agents were recorded for the *E. coli* ATCC 25922 reference strain (Figure 2; inoculation level: 1.80 × 10^6^ CFU/mL).

Once again, cinnamon and clove EOs were the most effective against *E. coli* because they kill most rapidly, that is, within 30 min after inoculation (after 15 min, only 1.8–2.0 × 10^5^ CFU/mL were recorded). In addition, lavender flower EO at 0.5% *v*/*v* killed quickly, and a complete reduction was noted after 1 h of incubation. Similar efficacy, although with lower dynamics, was found for a blend consisting of cinnamon EO at 0.0625% *v*/*v* (1/4 MIC) and enrofloxacin at 0.008 µg/mL (1/2 MIC); an additive effect in the checkerboard method was noted for this blend. Enrofloxacin at the MIC (0.016 µg/mL) required 4 h to completely eliminate *E. coli* ATCC 25922. The second blend (enrofloxacin at 1/2 MIC mixed with only 1/8 MIC of lavender EO) was characterised by identical time-to-kill values and similar dynamics. The last blend, consisting of 1/2 MIC of enrofloxacin and 1/4 MIC of clove, had the slowest activity; it required 6 h to completely inactivate this strain. Single components of the three blends (lines (8) to (11)) administered individually in the first 6 h of incubation were characterised by bacteriostatic activity, followed by the initiation of logarithmic growth, typical of *E. coli*. In contrast, control samples (lines (12) and (13)) were only characterised by logarithmic growth (up to 2.85 × 10^10^ CFU/mL). The same graph, but on a full logarithmic scale, is available as Appendix A.

## 3. Discussion

Natural plant products (e.g., essential oils) are important sources of novel therapeutic molecules and have various applications; however, they are mainly used in the cosmetic and food industries [16]. Moreover, these small molecules, alone and in combination, have powerful antiseptic, anti-inflammatory, antibacterial, antioxidative, and immune-boosting properties [17]. Usually, the major component (determining the chemotype of the EO) reflects the biophysical and biological features of the essential oil from which they were isolated. In addition, their mode of action depends on their concentration and when tested alone or in combination with other antimicrobial agents [18]. Unfortunately, new classes of antibiotics have appeared sporadically for the past 10 years, and most large pharmaceutical companies have left the field of new antibiotics or other antimicrobial agents. This task is now chiefly undertaken by academic laboratories and small-to-medium-sized companies [19].

The effects of cinnamon and clove EOs are independent of the degree of antibiotic resistance in *E. coli*. Because of the large number of constituents, in contrast to antibiotics, EOs seem to have no single specific cellular targets, but they “attack” comprehensively, by destroying the structure of the cell membrane, general leakage of the bacterial cell contents, and reducing the expression of certain genes [20]. However, the question is whether their biological effects are the result of the synergism of all molecules or reflect only those of the major molecules present at the highest levels. However, in the case of lavender EO, a relationship was observed between the decrease in antibiotic resistance and the two-fold stronger effect of this EO, which is in agreement with previous observations by Adaszyńska-Skwirzyńska et al. [21].

The genus *Cinnamomum* comprises hundreds of species belonging to the *Lauraceae* family, which are distributed throughout Asia and Australia. *Cinnamomum zeylanicum* Blume (also known as *Cinnamomum verum* J. Presl) is an indigenous tree of Sri Lanka (Ceylon), the true source of cinnamon bark and leaf essential oils [22]. Several studies have reported that (E)-cinnamaldehyde (also known as trans-cinnamaldehyde) is the major chemical compound of *C. zeylanicum* bark essential oil (55–78%) and contains only approximately 1–5% eugenol (as well as 1–5% of each other significant compounds like linalool, cinnamyl acetate, ß-Caryophyllene or 1,8-Cineol) while eugenol (60–80%) is the main compound in EO that is extracted from leaves [23,24] thus cinnamon leaf EO mimic clove bud EO which is also included in this manuscript. The Plant Therapy^®^ EO chemotype cinnamaldehyde (73.6%) tested in the present study met all the criteria for cinnamon bark essential oil. The spicy taste and fragrance of cinnamon are due to the presence of cinnamaldehyde, which is produced by the absorption of oxygen. As cinnamon bark “matures”, it goes dark, improving the resinous compounds [25]. In addition to being used as a spice and flavouring agent, cinnamon has been used as an anti-inflammatory, nematocidal, larvicidal, insecticidal, antimycotic and anticancer agent [26]. Due to current restrictions on antibiotics in chicken production, the poultry industry has looked towards novel alternatives. Dietary supplementation of poultry feed with cinnamon as a natural feed additive has beneficial effects on nutrient digestibility, immunity, blood biochemical profile and particularly on gut health to alleviate the impact of disease and heat stress by maintaining water and electrolyte balance and feed intake [27]. In addition, cinnamon essential oil resulted in an acceptable level of virulence gene downregulation in poultry respiratory bacterial agents, including the *Escherichia coli stx1* gene [28]. Unfortunately, this oil belongs to the “hot” EOs group. Therefore, the maximum content should not exceed 0.1% for topical application (manufacturer’s recommendations = 1.03 mg/mL). However, oral administration has not yet been well described, especially in poultry. A study performed by Chowdhury et al. [29] suggested that cinnamon EO at 0.3 g/kg broiler diet could lower pathogenic bacteria (*E. coli* and *Clostridium* sp.) in the intestine and improve gut morphology along with improvement of immune response. Pure *trans*-cinnamaldehyde was administrated to broilers in drinking water at 0.06% fully inactivated *Salmonella* after 24 h [30]. The in vitro antibacterial activity of the cinnamon bark EO against *E. coli* was also investigated. Alizadeh Behbahani et al. [31] used a hydrodistillation extraction technique to obtain cinnamon oil from the dried bark (71.50% of (E)-cinnamaldehyde and linalool—7.00%, β-caryophyllene—6.40%, eucalyptol—5.40% and eugenol—4.60% as the main components). MIC for *E. coli* ATCC 25922 was 6.25 mg/mL (for comparison, 2.56 mg/mL in our study). Stronger effects of this oil on Gram-positive bacteria have been reported. Results similar to our study (MIC 2.5 mg/mL) were obtained by Raeisi et al. [32] using an essential oil obtained by hydrodistillation of the bark of local cinnamon (*E. coli* ATCC 43894; main components: cinnamaldehyde—79.74%, trans-calamenene—2.62%, benzaldehyde—1.71%, borneol—1.73%, cinnamyl acetate—1.58%) and Ebani et al. [33] using commercial FLORA^®^ EO (Pisa, Italy; 56.4% of (E)-Cinnamaldehyde and β-caryophyllene—10.3%) and an *E. coli* strain, isolated in a case of poultry colibacillosis. However, other studies have reported a lower MIC for cinnamon EO: 1 mg/mL (92.4% cinnamaldehyde; *E. coli* ATCC 25922) [34] and 0.625–2.5 mg/mL (chemical composition unknown; *E*. *coli* O157:H7) [35]. Nemattalab et al. [36] reported that the MIC of cinnamon EO (97.44% of (E)-cinnamaldehyde; origin of EO unknown) ranged between 155 and 165 µg/mL, which is in agreement with the results of Lu et al. [37] (MIC of 100–400 µg/mL). Surprisingly, El Atki et al. [38] reported a much lower MIC (only 4.88 µg/mL) for cinnamon EO from Chinese cinnamon bark (*Cinnamomum cassia*) against *E. coli* 25922. Only a few studies have investigated the activity of cinnamon oil against APEC serotypes in poultry. A total of 117 *E. coli* APEC strains (serotypes O78, O2 as well as O128 and O139) and commercial cinnamon EO (Erba Vita, San Marino; 88.2% cinnamaldehyde) were used for the analysis by Casalino et al. [39]. Treatment with ≥1 mg/mL of cinnamon EO was effective against all APEC serotypes, regardless of the bacterial cell density used in the experiments (up to 10^8^ CFU/mL). Identical results were obtained by Van et al. [40] for 10 field isolates (commercial Heber EO, Vietnam; 91.9% of cinnamaldehyde). Cui et al. [41] have also tested cinnamon oil (bought from J.E International, France) on *E*. *coli* ATCC 25922. The main components were eugenol (75.5%) and eugenyl acetate (4.4%). Both components suggested that the EO was derived from cinnamon leaves (not bark). Therefore, their MIC (0.05% *v*/*v*) should correspond to the activity of clove oil (in our study—0.125% *v*/*v*) instead of the conventional cinnamon bark EO (in our study—0.25% *v*/*v*).

*Eugenia caryophyllus* (Spreng) Bullock and S. G. Harrison (syn. *Syzygium aromaticum* (L.) Merr., *Myrtaceae*) is a tropical tree whose buds provide a source of essential oil that is widely applied in dental medicine and cosmetics [42]. The basic constituents of EO are eugenol (at least 50%; however, more often 75–88%) and the remaining constituents mainly consist of eugenyl acetate, ß-caryophyllene, and α-humulene [43]. In clove bud EO, the second most significant component is usually eugenyl acetate (8.61–21.32%), while in the leaf and stem, it was detected in considerably lower amounts (0–1.45% and 0.07–2.53%, respectively). In leaf essential oils, the second main compound is β-caryophyllene (11.65–19.53%) which is less represented in bud EOs (2.76–8.64%) and stem essential oils (1.66–9.72%) [44]. The Plant Therapy^®^ EO chemotype eugenol (81.4%) supported by eugenyl acetate (9.5%) tested in the present study met all criteria for clove bud essential oil. Clove oil is less ‘aggressive’ than cinnamon oil and, according to manufacturers’ recommendations, the maximum content should not exceed 0.5% for topical application. The antibacterial activity of clove flower EO (eugenol—79.8%, eugenyl acetate—9.6%, ß-caryophyllene—7%) against extended-spectrum β-lactamase-producing bacteria (including *E. coli* isolates originating from chicken meat in traditional markets and *E. coli* ATCC 25922) was the subject of research by Ginting et al. [45]. The MIC was 0.078% *v*/*v* for all the strains tested (0.125% *v*/*v* in our study, regardless of the degree of antibiotic resistance). Clove bud EO (obtained by steam distillation from buds at the mature stage; 78.55% of eugenol, 15.75% of ß-caryophyllene and 4.28% of humulene) was effective against *E. coli* ATCC 25922 at 0.64 mg/mL (for comparison, 1.31 mg/mL in our study). The MIC for pure eugenol was lower—0.32 mg/mL [46]. The antimicrobial potential of clove EO (eugenol 77.32%, caryophyllene 16.77% obtained by hydrodistillation of dried flower buds) was investigated in 135 clinical isolates (from human urinary and gastrointestinal tract) of extended-spectrum β-lactamase-ESBL-producing *Escherichia coli* using the broth dilution method [47]. High MICs were obtained, usually 20 mg/mL. MIC results, which are directly dependent on eugenol content, were reported by Sohilait et al. [48] for *Escherichia coli* FNCC0091. This study revealed MIC values for clove steam (eugenol 97.75%) and leaves (eugenol 82.97%) at ≥2.5 µL/mL (approx. 0.25% *v*/*v*) and ≥5 µL/mL (approx. 0.5% *v*/*v*) for buds (eugenol 75.3%). One APEC and two non-APEC chicken strains were tested by Kammon et al. [49] using clove EO (1.04 g/mL, unknown content, BDH Laboratory Supplies, England). MICs recorded in mg/mL were 3.12–6.25 (approx. 0.3125% to 0.625% *v*/*v*). Commercial EO from clove provided by Pollena-Aroma (Nowy Dwór Mazowiecki, Poland; eugenol 86%, ß-caryophyllene 9.8%) was analysed by Dąbrowski et al. [50] on 30 clinical isolates obtained from the urine of patients with urinary tract infections (UTI). MIC ranged from 2.1 to 3.1 mg/mL (2.6 mg/mL in more than 70% of cases). In contrast, Faujdar et al. [51] reported a constant MIC of 0.39 mg/mL for *E. coli* ATCC 25922 as well as 32 ESBL-producing UTI strains, 18 AmpC-beta-lactamase-producing UTI strains and 50 non-(ESBL, AmpC, and metallo-beta-lactamases (MBL)) producing UTI strains, regardless of its antibiotic resistance. It should be mentioned that 100 mg/mL bud extract (not EO, contents unknown) was used in this study. However, the most commonly tested activity of this oil is against the *E. coli* O157:H7 serotype. In the study by Yoo et al. [52], using commercial Now Food clove EO (extracted by steam distillation of clove buds, leaves, and stems; 88.9% of eugenol), the MIC against *E. coli* O157:H7 (strain NCCP 15739) was 0.05% v/v. In contrast, when using broth culture medium (in tubes), the MIC for *E. coli* O157:H7 (strain unknown) was determined to be 6 mg/mL (approx. 0.6% *v*/*v* of clove EO extracted from clove powder with 86.04% eugenol) [53]. The antimicrobial activity of clove and cinnamon EOs (unknown contents) against 26 foodborne pathogens (incl. six different strains of *E. coli* O157:H7) was analysed by Hoque et al. [54]. Both EOs had a similar effect with an MIC of 2.5 mg/L (in our study on APEC 1.31 mg/mL and similar 2.56 mg/mL, respectively). In addition, cinnamon bark and clove bud EOs inhibited Enterohemorrhagic *Escherichia coli* O157:H7 (EHEC) biofilm formation by more than 75% and eugenol reduced fimbriae formation by EHEC and downregulated several virulence genes [55].

The *Lamiaceae* family and *Lavandula* genus contain many aromatic and medicinal plants of which *Lavandula angustifolia* Mill. is the best-known source of lavender essential oils. *L. angustifolia* is extensively cultivated in some countries, especially Bulgaria, France, Greece, the United Kingdom, Spain and Morocco. True lavender EO is highly valued because of its attractive fragrance in comparison to spike oil (from *L. latifolia*) and lavandin oil (from *L. x intermedia*) [56]. True lavender EO is characterised by a high content of linalool and linalyl acetate (both at a similar level, approx. 20–45%), a moderate amount (0.5–8%) of lavandulyl acetate, lavandulol and terpinene-4-ol, and variable levels of eucalyptol and camphor [57,58,59]. Once again, the Plant Therapy^®^ EO chemotype linalyl acetate (31.74%)/linalool (27.62%) tested in the present study met all the criteria for lavender essential oil. Lavender essential oil belongs to the group of weak allergens, and according to manufacturers’ recommendations, the maximum content should not exceed 2% (optimum) to 5% (maximum) for topical application. Although lavender oil is quite popular, there are no studies on its activity against poultry isolates. One of the limited studies in this area is the research by Adaszyńska-Skwirzyńska et al. [21] using a commercial Avicenna EO (Wrocław, Poland). The main ingredients were linalool 35.17% and linalool acetate 46.25%. Similar to our study, for the five broiler field isolates (however, serotypes were not specified), the MIC was 1% *v*/*v* and for the reference strain *E. coli* ATCC 25922 0.5% *v*/*v*. The reference strain, *E. coli* ATCC 25922, was tested more frequently. For this strain, Puvaća et al. [60] report a lower MIC of 2.1 mg/mL (4.4 mg/mL in our study). However, this commercially available essential oil purchased from a local distributor in Novi Sad (Serbia) had an atypical composition: carbitol (13.05%) and α-terpinyl acetate (10.93%) followed by linalool (10.71%) and linalyl acetate (9.6%). *Lavandula angustifolia* Sevastopolis EO (Romania) from the study by Predoi et al. [61] also had a rather unusual composition: high linalool content (47.55%) with low levels of linalyl acetate (only 3.75%; there was more camphor (9.67%), 1,8-cineole (8.6%), borneol (8.52%), and terpinene-4-ol (3.8%)). *E. coli* ATCC 25922 and *E. coli* ESBL 4493 were susceptible, with MIC values ranging from 0.1% *v*/*v* to 0.19% *v*/*v*, indicating strong antimicrobial activity of this lavender EO. *E. coli* O157:H7 cells treated for 2 h with pure linalool and observed by scanning electron microscopy have shown significant structural changes [62]. Linalool, after contact with a bacterial cell, first acts on the cell membrane resulting in reduced membrane potential and structure, followed by intracellular leakage of macromolecules (DNA, RNA and proteins). In addition, it inhibits energy-related pathways and the activity of key enzymes, as comprehensively described in a review by Mączka et al. [63]. Linalyl acetate also showed antimicrobial properties against *E. coli* ATCC 15221 with an MIC of 5.0 mg/mL [64]. There is increasing evidence that the antimicrobial activity of lavender oil is dependent on the country of origin and chemotype; for example, the Bulgarian-type (51.9% linalool, 9.5% linalyl acetate) was effective against 23 of 25 bacteria, whereas the French-type (43.2% linalyl acetate, 29.1% linalool) was only effective against 13 bacteria [65]. This suggests that linalool (alone or, more likely, in synergy with other ingredients) rather than linalyl acetate determines the activity of the lavender essential oil.

It is known that serotypes O78, as well as O1 or O2, are commonly associated with infections in chickens (more than 80% of the cases) [66]. In our study, similar results were obtained (75% of the isolates). Broiler chicks undergo stress from hatching to their placement on the farm. Each placement of chicks may result in the introduction of different APEC serotypes with unknown drug susceptibilities. Undiagnosed treatment is often ineffective. Such APEC serotypes can survive until the end of rearing and can be detected in broiler meat after slaughter. Moreover, resistance to antibiotics may increase over time. In a study by van der Horst et al. [67], the acquisition of resistance to amoxicillin, tetracycline, and enrofloxacin by *E. coli* was tested by exposing living cells to constant or stepwise increasing concentrations of these compounds. The MIC for enrofloxacin increased from 0.25 µg/mL (upper sensitivity limit) to maximally 512 µg/mL (which is significantly higher than our extremely resistant MDR-1 isolate with an MIC of 64 µg/mL) after two weeks of exposure to low concentrations of enrofloxacin. The origin of the MDR-3 serotype O1 isolate resistance is well known to the first author of the present study. After the first reported mortalities, one of the broiler breeder flocks started treatment with enrofloxacin, but without success. Doxycycline was administrated after a further two weeks, followed by amoxicillin with clavulanic acid. On each occasion, swabs were taken from the internal organs and *E. coli* with increasing resistance were isolated. As it later turned out after real-time PCR testing, the primary cause of the problems in the herd was infectious bronchitis virus (IBv). The consequence of this situation was an outbreak of the MDR-3 isolate in 1-day-old chicks originating from this flock. The use of herbs, spices and increasingly, in the light of our research, essential oils during the rearing process can prevent this complex phenomenon. The review encompasses recent studies regarding the protection against pathogenic *E. coli* by EO with a major focus on the inhibition of toxins and proliferation in food is well described by Munekata et al. [68]. The ability to disrupt the membrane of *E. coli* cells and facilitate intracellular compound leakage is well documented in scanning electron microscopy images of both cinnamon EO [31,41] and clove EO [45,52]. Unfortunately, the use of EO has some disadvantages. First, an intense specific fragrance (even at a dilution of 0.1% *v*/*v*), a bitter taste (e.g., eugenol), poor water solubility, high volatility and low stability (e.g., linalool from lavender) may limit the possibility of its use on the farm (in drinking water, feed or as aromatherapy). Second, the intense scent/taste may be a direct result of the MIC, which is usually up to 1000x higher than the MIC for antibiotics, making antibiotics easier and cost-effective to administer in an effective dose compared to EOs. Therefore, it is preferable to exploit the synergy between EOs and antibiotics. However, there is still relatively limited research in this field. El Atki et al. [38] reported a synergy of cinnamon EO with chloramphenicol (FICi = 0.5) and an additive effect with streptomycin (FICi = 1.0) in *E. coli* ATCC 25922. Similar to our study, Adaszyńska-Skwirzyńska et al. [21] suggest a high potency of lavender EO to interact with enrofloxacin: additive effect to *E. coli* ATCC 25922, susceptible and intermediate strains (the FIC index between 0.56 to 1.0) and synergy in regard to enrofloxacin-resistant field strains. In our study, we also found that additional synergy is possible between enrofloxacin and cinnamon EO or clove EO. To our knowledge, this is the first study to analyse the effect of cinnamon and clove EOs on enrofloxacin used to control APEC strains in poultry. The synergy between other antibiotics and essential oils has been well documented [69,70,71].

Enrofloxacin is a chemotherapeutic (not an antibiotic *sensu stricto*) that belongs to the fluoroquinolone group. It was synthesised in 1983 from nalidixic acid; however, the first product was released in 1991 as an oral drug for poultry under the trade name Baytril^®^ (Bayer, Germany). The molecular targets of enrofloxacin are enzymes that control DNA topology: gyrase and topoisomerase IV [72]. The natural consequence of this process is the inhibition of bacterial DNA replication. Moreover, enrofloxacin is not approved for use as a drug in humans. Antibiotic resistance can be acquired through three main mechanisms: (1) transfer of resistance genes from resistant to susceptible microorganisms; (2) genetic adaptation, and (3) phenotypic adaptation, which primarily increases the expression of existing cellular machinery such as efflux pumps [73]. Multidrug-resistant APEC strains present in poultry products (meat, eggs, etc.) may potentiate the first mechanism because bacteria can share their genes with each other in a process called horizontal gene transfer. This can occur between bacteria of the same species or between different species on the path of conjugation, transduction or transformation [73]. It can affect not only *E. coli* but also other enteric pathogens causing food poisoning, such as *Salmonella* spp., *Staphylococcus aureus*, *Campylobacter* sp. or human pathogens that acquire resistance genes. This can be a risk to consumers of poultry products if they are not properly processed.

The positive interaction between enrofloxacin and essential oils (synergy or additive effect) has not yet been sufficiently established. The development of resistance to fluoroquinolones occurs in several ways. The first is the presence of different quinolone resistance (*qnr*) genes in *E. coli* plasmids [74], which are capable of protecting the target gyrase and topoisomerase. Leakage of macromolecules (including plasmids) after cinnamon, clove or lavender “strike” may reduce the protective potential and enrofloxacin becomes more active than the individual MIC for enrofloxacin might suggest. Second, efflux pump systems are present. The efflux pump system decreases the intracellular concentration of fluoroquinolones by transporting, for example, enrofloxacin from the cell to the environment [72]. As mentioned earlier, all the essential oils studied in the manuscript significantly damaged the structure of the membrane and thus could significantly inactivate the pumps. Enrofloxacin can act more effectively against *E. coli* than the initial MIC implies. Third, there is the presence of a gene encoding the aminoglycoside acetyltransferase AAC(60)-Ib-cr (also within the plasmid), an enzyme that modifies fluoroquinolones by acetylation [75]. Once again, resistance is conditioned by the presence of plasmids making it vulnerable to changes in cell structure induced by essential oils. Finally, mutations appear in the quinolone resistance determinant region (QRDR) within the subunits forming topoisomerases II and IV. The occurrence of some mutations leads to abnormal conformation of the subunits and reduced binding affinity of, for example, enrofloxacin to the DNA-gyrase or DNA-topoisomerase IV complex [72]. Probably, chromosome-borne mechanisms are the most resistant to essential oil activity. This may explain the observed fact that isolates with the highest MIC for enrofloxacin (MIC > 16 µg/mL; MDR-1 to MDR-3) are still classified as resistant to this antimicrobial despite the observed interaction. Unfortunately, the genetic basis of enrofloxacin resistance in all isolates under study has not been determined.

It is important to emphasize that the administration of an essential oil with enrofloxacin does not have a strict therapeutic purpose, because this is still what antibiotics are for. EO is intended to initiate damage to bacterial cells, facilitating the activity of enrofloxacin and, consequently, preventing the emergence of increasing resistance to this antimicrobial agent. The effectiveness of the EOs was confirmed by time–kill curve analysis.

The dynamics of essential oil and/or enrofloxacin activity cannot be assessed during checkerboard incubation. Visual reading only occurred at the end of the incubation period (up to 24 h). Time–kill studies have shown an extremely fast activity of cinnamon and clove oils (up to 1 h), as well as the fast effect of lavender oil (up to 4 h). Blends with a lower than MIC concentration of enrofloxacin mixed with a lower EO content (usually 1/4 MIC) required 6 ± 2 h to achieve a similar effect. Information related to other similar studies is very limited, especially for APEC strains.

A study by Iseppi et al. [76] was to assess the efficacy and synergistic potential of two essential oils (cinnamon and clove) traditionally used in the food industry to control food-borne pathogens in fresh-cut fruits (including *E. coli* ATCC 25922). Both singles (MIC for cinnamon at 8 µg/mL and 4 µg/mL for clove) and a blend consisting of these oils showed a reduction in viable *E. coli* ATCC 25922 cells of about 2 log CFU/g after 24 h. At the end of the trial (8 days), the EO/EO combination had the best results (reduction by 7.7 log CFU/g *E. coli* viable cells) followed by single EOs (reduction by approx. 6 log CFU/g). It should be noted, however, that the initial number of bacteria was higher than that in our experiment—approx. 10^8^ CFU/g and EOs content multiple times lower than our MICs. A study by Yap et al. [11] investigated the mechanism of action of cinnamon bark EO (MIC at only 0.02% *v*/*v*) when used singly and in combination with piperacillin, for its antimicrobial and synergistic activity against the well-described ß-lactamase TEM-1 plasmid-conferred *Escherichia coli* J53 R1 strain. Similar to our study, the single components of the blend were ineffective, and cultures proceeded to unlimited logarithmic growth of viable cells of this bacterium over a period of 4–8 h of incubation; however, the blend itself was bactericidal after 20 h of incubation, meaning synergy has been confirmed. The time–kill curve assays revealed the occurrence of bactericide synergism in combinations of *C. zeylanicum* bark (0.25 mg/mL; 1/10 of our MIC) with rosemary [77]. At this very low concentration of cinnamon EO, a bacteriostatic effect of single cinnamon EO on *E. coli* ATCC 25922 was noted for the first 12 h and a bactericidal effect after 24 h. In addition, after 24 h incubation, the synergistic effect of cinnamon bark EO (MIC at 0.8 µg/mL) or cinnamaldehyde (MIC at 0.15 mg/mL) with gentamicin against ESBL-producing *E. coli* isolates and ATCC 25922 reference strain was confirmed by time–kill curve experiments [78]. Again, the individual components were ineffective when used individually. Pure eugenol (MIC 0.25 mg/mL) was sufficient to fully eradicate *E. coli* strain 128 MR within 2 h whereas 1/2 MIC had only a bacteriostatic effect [79]. In the study of Wang et al. [80], for most rare clinical colistin-resistant or native colistin-sensitive *E. coli* strains as well as the ATCC 25922 reference strain, eugenol exhibited a synergistic effect (FICi from 0.375 to 0.5) or additive effect (FICi = 0.625) with colistin and a bactericidal effect within 2 h in the time–kill assay was noted. The mode of action of lavender EO on antimicrobial activity against multidrug-resistant *Escherichia coli* J53 R1 strain (carrying a plasmid encoding beta-lactamase TEM-1) when used singly and in combination with piperacillin was studied by Yap et al. [81]. In their time–kill analysis, the complete killing of this bacterium was observed within 4 h when lavender EO (0.5% *v*/*v*; MIC similar to that in our study) was combined with piperacillin. Lavender EO and piperacillin administered alone at sub-concentrations did not show a complete killing profile within the time of the study.

## 4. Materials and Methods

### 4.1. Escherichia coli Strains

Sixteen field isolates of *Escherichia coli* (isolated from the hearts and yolk sacs of 1-day-old broilers that died during transport from 2017 to 2021) were tested. All field isolates were retrieved from a frozen strain bank (live animals were not included in the experiment). The strain collection included ten multidrug-resistant strains (resistance to enrofloxacin and more than three other antibiotic groups but susceptibility to colistin; labelled as MDR 1–10), three strains resistant to various antimicrobial groups (≤3) but known intermediate resistance to enrofloxacin (SDR 1–3) and three *E. coli* strains sensitive to all antimicrobials tested (SENS 1–3). Additionally, a non-APEC and antibiotic-sensitive reference strain of *E. coli* ATCC 25922 (WDCM 00013; serotype O6; KWIK-STIK™ Plus, Microbiologics, St. Cloud, MN, USA) was used. ATCC 25922, originally isolated from a human clinical sample in the USA, is the recommended reference strain for antibiotic susceptibility and media testing. APEC affinity was tested using diagnostic sera (Sifin Diagnostics GMbH, Berlin, Germany) according to the manufacturer’s recommendations. A positive result was confirmed in the Widal reaction (microtitre plate confirmation test) to exclude the effects of any parallel nonspecific agglutination. The following diagnostic sera were used: polyspecific Anti-coli A (preliminary recognition of APEC) and monospecific (O1, O2, O18, O78). However, the ability to produce toxins is unknown. Drug resistance results were compiled from official test reports. Before inoculation on a checkerboard, each strain was revived on Columbia agar with the addition of 5% sheep blood (Graso, Starogard Gdański, Poland) and incubated for 24 h at +37 °C ± 1 °C.

### 4.2. Antimicrobial Agents

Cinnamon bark oil (from *Cinnamomum zeylanicum*; origin Sri Lanka), clove bud oil (from *Eugenia caryophyllus*; origin Indonesia) and lavender flower oil (from *Lavandula angustifolia*; origin Greece) were used to assess the sensitivity of the above-mentioned *E. coli* strains to essential oils. The essential oils were purchased from Plant Therapy^®^ (Twin Falls, ID, USA). The density of the essential oil was assessed by weighing 1 mL of each oil (mean from 10 replicates). The manufacturer provides gas chromatography-mass spectrometry (GC-MS) reports for individual lots of essential oils (on the official website or on request). In addition, enrofloxacin (100 mg/mL; Medoxil Oral, Medivet S.A., Śrem, Poland), an antimicrobial agent belonging to the fluoroquinolone group, was also used. This ready-to-use solution can be administered to chickens in drinking water. It also contained benzyl alcohol (7.5 mg/mL) as an auxiliary substance. The essential oils were diluted in acetonitrile (LiChrosolv^®^, Supelco, Merck KGaH, Darmstadt, Germany) to create a gradient ranging from 10% to 0.01% *v*/*v*. Enrofloxacin was pre-diluted in sterile 0.9% saline (Ecotainer^®^, B. Braun Medical AG, Sempach, Switzerland) creating a gradient from 5.12 mg/mL to 0.01 µg/mL. This allowed the selection of appropriate serial dilutions for resistant, intermediate and sensitive strains during the construction of checkerboards.

### 4.3. Gas Chromatography-Mas Spectrometry (GC-MS) Reports

According to the manufacturer’s official GC-MS reports, all essential oils purchased from Plant Therapy^®^ can be considered flagship examples in their category. Cinnamon bark essential oil (LOT: CC0109R) is a typical member of the cinnamaldehyde chemotype (73.64%) followed by eugenol (3.41%), 1,8-cineole (3.39%), cinnamyl acetate (2.77%), linalool (2.47%), α-pinene (2.45%), ß-caryophyllene (2.0%), limonene (1.35%), p-cymene (1.04%), α-phellandrene (0.86%), α-terpinene (0.57%), (Z)-cinnamal (0.43%), benzyl benzoate (0.38%), α-humulene (0.34%), α-terpineol (0.31%), camphene (0.28%), β-pinene (0.26%), hydrocinnamal (0.25%), thujene (0.24%), terpinen-4-ol (0.23%), α-copaene (0.19%), caryophyllene oxide (0.17%), benzaldehyde (0.17%) and camphor (0.11%). Approximately 70 other components were also found in trace amounts (≤0.1%). Clove bud essential oil (LOT: CG0108R) is an example of the eugenol chemotype (81.41%). It also contained eugenyl acetate and ß-caryophyllene in smaller quantities (9.48% and 6.32%, respectively) as well as α-Humulene (0.75%), caryophyllene oxide (0.34%) and α-copaene (0.16%). Trace amounts of the remaining 35 compounds were present. The last essential oil that was used, lavender flower essential oil (LOT: L50115R), contained the predominant percentage of linalyl acetate (31.74%) and linalool (27.62%) as well as a variety of other components such as the following: (Z)-β-ocimene (8.03%), terpinen-4-ol (5.65%), lavandulyl acetate (3.69%), (E)-β-ocimene (3.61%), β-caryophyllene (3.28%), (E)-β-farnesene (2.64%), α-terpineol (1.02%), octen-3-yl acetate (0.99%), lavandulol (0.78%), octan-3-one (0.74%), myrcene (0.65%), 1,8-cineole (0.60%), hexyl acetate (0.49%), geranyl acetate (0.44%), borneol (0.41%), α-santalene (0.41%), geraniol (0.38%), α-pinene (0.29%), limonene (0.27%), neryl acetate (0.27%), camphor (0.22%), γ-terpinene (0.21%), octen-3-ol (0.21%), p-cymene (0.17%), nerol (0.16%), α-thujene (0.14%), camphene (0.13%) and bornyl acetate (0.13%). In addition, this oil contained more than 70 trace amounts of other components. Even after dilution of the base oil, each of these components can determine the activity of the EO, especially the interaction with the primary chemotype-determining components (cinnamaldehyde, eugenol, linalyl acetate/linalool) or an additional antimicrobial agent (such as enrofloxacin).

### 4.4. Checkerboards

The creation of so-called checkerboards enabled the simultaneous estimation of individual minimum inhibitory concentration (MIC) for each antimicrobial agent as well as the determination of interactions between the selected essential oil and enrofloxacin (three possible combinations per bacterium: cinnamon × enrofloxacin, clove × enrofloxacin and lavender × enrofloxacin). Checkerboards were prepared in 96-well plates with a cover (Wuxi Nest Biotechnology, Wuxi, China). For the growth medium, 170 µL of Mueller–Hinton broth (MHB) (GRASO, Gdansk, Poland) was initially added to each well. Horizontal gradients of enrofloxacin were then performed (20 µL of each two-fold consecutive dilution—a total of ten columns; 1:10). The eleventh column did not contain enrofloxacin, only saline in identical proportions. The complementary gradient of essential oil was made vertically (10 µL of each of seven two-fold consecutive dilutions, the last eight rows did not contain EO but only acetonitrile in analogous proportions; 1:20). As high concentrations of acetonitrile (≥10%) can inhibit the growth of Gram-negative rods (own observations), it is important that the final concentration of acetonitrile in the well should not be higher than 5%. Such a situation occurs, inter alia, in the last twelfth column, reserved for controls. In this area of the checkerboard, the wells contained only MHB, saline and acetonitrile (purity/negative control), while bacteria were added to only half of them (growth-positive control). At the end, a bacterial suspension with a final concentration of approximately 1.5 × 10^6^ colony forming units (CFU) per well (derived from 0.5 McFarland) was added simultaneously at a ratio of 1:10 to the 92 wells (excluding the four wells intended as negative controls). To prevent the transfer of bacteria between wells during incubation and the loss of some of the culture volume (caused, among other things, by evaporation), the entire plate was tightly covered with a protective breathable film (Axygen™, Thermo Fisher Scientific, Waltham, MA, USA). The plates were incubated for 18 h at +37 °C ± 1 °C. Each checkerboard test was performed in triplicate.

After incubation, owing to the possibility of false turbidity or sediment formation originating from essential oil at higher concentrations (≥0.1%), false results may be obtained. To detect the presence of viable bacterial cells in each well, 20 µL of 0.01% resazurin (POL-AURA, Olsztyn, Poland) was added to each well after incubation, sealed, and incubated for an additional 6 h (maintaining sterility is crucial). Resazurin is dark blue but changes colour to various shades of pink in the presence of live cells. The intensity of the pink colour was directly proportional to the number of live cells that originally survived the first incubation in the presence of single or both antimicrobial agents. In this case, the MIC was the lowest concentration of an antibacterial agent expressed in μg/mL (enrofloxacin) or % *v*/*v* (essential oils) which completely prevented the colour change (i.e., the last blue colour of the well which remained intact).

### 4.5. Interaction between Essential Oils and Enrofloxacin

To determine possible interactions between the three essential oils (cinnamon, clove and lavender) and enrofloxacin, the fractional inhibitory concentration (FIC) was calculated according to van Vuuren and Viljoen [82] using the following formulas: FIC_ENRxEO_ = MIC_ENRxEO_/MIC_ENR_ (reading in columns) and FIC_EOxENR_ = MIC_EOxENR_/MIC_EO_ (reading in rows) where: ENRxEO—MIC of enrofloxacin in the presence of essential oil; EOxENR—MIC of essential oil in the presence of enrofloxacin; EO—essential oil acting independently; ENR—enrofloxacin acting independently. The FIC index (FICi) was then calculated for each bacterial strain as the sum of the FIC: FICi = FIC_ENRxEO_ + FIC_EOxENR_. The FIC index is expressed as the interaction of two antimicrobial agents where the concentration of each test agent in combination is expressed as a fraction of the concentration (corresponding to 1/2 MIC, 1/4 MIC, 1/8 MIC, etc.) that would produce the same effect when used independently. The interpretation of possible in vitro interactions between enrofloxacin and other antimicrobial agents (cinnamon, clove and lavender essential oils) was described as synergistic (FICi ≤ 0.5), additive (0.5 < FICi ≤ 1.0), noninteractive (1.0 < FICi ≤ 4.0), or antagonistic (FICi > 4.0).

### 4.6. Time–Kill Analysis

*E. coli* ATCC 25922 and MDR-9 were challenged with essential oils and enrofloxacin at various concentrations and bacterial viability. This value was determined at different time points during the incubation period. The final concentrations of essential oils and enrofloxacin in MHB were as follows: cinnamon bark (0.25 % *v*/*v* as MIC and 0.0625 % *v*/*v* as 1/4 MIC—identical for both strains), clove bud (0.125 % *v*/*v* as MIC and 0.03125 % *v*/*v* as 1/4 MIC—identical for both strains), lavender (1% *v*/*v* as MIC and 0.25 % *v*/*v* as 1/4 MIC for MDR-9 and 0.5% *v*/*v* as MIC and 0.0625 % *v*/*v* as 1/8 MIC for ATCC 25922), enrofloxacin (2 µg/mL as MIC, 0.5 µg/mL as 1/4 MIC, 0.25 µg/mL as 1/8 MIC for MDR-9 and 0.016 µg/mL as MIC and 0.008 µg/mL as 1/2 MIC for ATCC 25922). In addition to mimicking synergy (MDR-9) or additive effect (ATCC 25922) conditions, three blends per strain of enrofloxacin with the respective essential oils were created as shown in Table 1. As controls, pure MHB and MHB with 5% acetonitrile were also tested. Each test tube contained a final volume of 10 mL. Immediately after incubation, viable cell counts were performed for 100 μL of the samples collected at 0 min (inoculation), 15 min, 0.5 h, 1 h, 2 h, 4 h, 6 h, 8 h, 12 h and 24 h. To quantify viable cells, a horizontal method was used to determine the number of *E. coli* according to the ISO 4833-1:2013 standard [83] with minor modifications. Briefly, immediately after collection, ten-fold serial dilutions of each sample were performed with 0.9% saline (Ecotainer^®^, B. Braun Medical AG, Sempach, Switzerland) on ice and 1 mL of each dilution was transferred to a Petri dish. Liquid Mueller–Hinton agar (Graso, Starogard Gdański, Poland) was then added and mixed gently. After complete solidification, plates were incubated at +30 °C for 72 h. After incubation, the colonies were counted manually. ISO 7218 [84] is the calculation method. The experiment was performed in triplicate.

## 5. Conclusions

An in vitro study of the antibacterial activity of essential oils showed that cinnamon (MIC of 0.25% *v*/*v*), clove (MIC of 0.125% *v*/*v*) and lavender EOs (MIC ranged 0.5–1% *v*/*v*) had acceptable antibacterial activity against *E. coli* isolated from broilers (including multidrug-resistant APEC strains), which make these antimicrobial agents a potential candidate for the treatment of *E. coli* infections. Lavender oil had the best and highest percentage of synergy cases with enrofloxacin (82.35%), although cinnamon and clove oils also had this desirable potential (synergy in 64.7% and 47.1% cases, respectively). In light of our time–kill study and other studies, it can be concluded that long-term administration of multiple lower doses of essential oils can be carried out than would result directly from the MICs. At the same time, these EOs have a high potential for synergism with antibiotics applied over several days to control APEC strains in chicks. These combinations can be used as alternative therapeutic applications, which could decrease the minimum effective dose of the drugs, thus reducing their possible adverse effects and the costs of treatment. It is also important to consider whether, by analogy with antibiotics, the long-term use of essential oils will result in the acquisition of stepwise resistance.

## Figures and Tables

**Figure 1 ijms-25-05220-f001:**
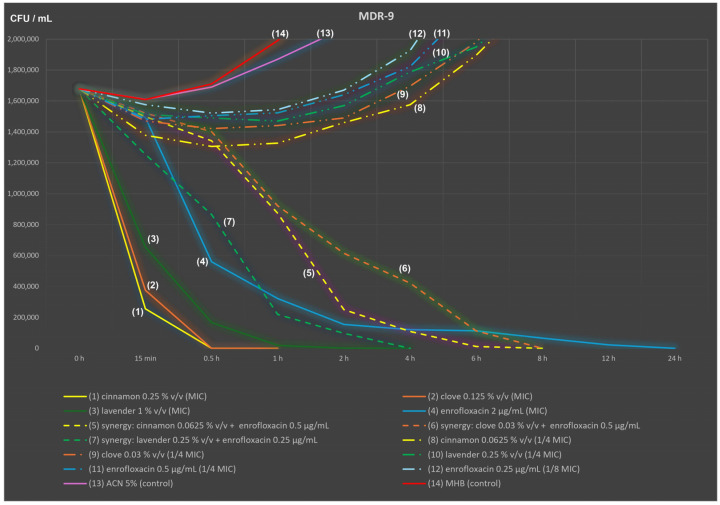
Time–kill analysis of cinnamon bark, clove bud and lavender flower essential oils administered alone and in the synergistic combination with enrofloxacin (*Escherichia coli* MDR-9 strain).

**Figure 2 ijms-25-05220-f002:**
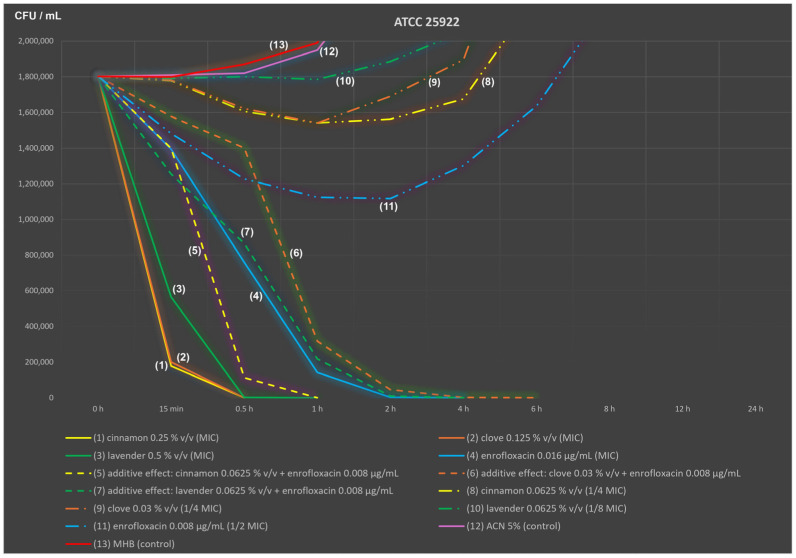
Time–kill analysis of cinnamon bark, clove bud and lavender flower essential oils administered alone and in combination with enrofloxacin (additive effect; *Escherichia coli* ATCC 25922 reference strain).

**Table 1 ijms-25-05220-t001:** *Escherichia coli* susceptibility test results for enrofloxacin and three essential oils (cinnamon, clove and lavender) and the estimation of interactions (best match within triplicates).

Strain [Serotype]	Antimicrobial Agent	Cinnamon				Clove				Lavender			
		MICi	MICc	FIC	FICi	MICi	MICc	FIC	FICi	MICi	MICc	FIC	FICi
MDR-1 [O78]	ENR (µg/mL)	64	16	0.25	0.5 SYN	64	16	0.25	0.75 ADD	64	16	0.25	0.28 SYN
	EO (% *v*/*v*)	0.25	0.0625	0.25		0.125	0.06	0.5		1	0.03	0.03	
MDR-2 [O78]	ENR (µg/mL)	16	4	0.25	0.5 SYN	16	2	0.125	0.625 ADD	16	4	0.25	0.5 SYN
	EO (% *v*/*v*)	0.25	0.0625	0.25		0.125	0.0625	0.5		1	0.25	0.25	
MDR-3 [O1]	ENR (µg/mL)	16	4	0.25	0.75 ADD	16	8	0.5	1 ADD	16	1	0.06	0.56 ADD
	EO (% *v*/*v*)	0.25	0.125	0.5		0.125	0.0625	0.5		1	0.5	0.5	
MDR-4 [O78]	ENR (µg/mL)	8	1	0.125	0.625 ADD	8	0.25	0.03	0.53 ADD	8	1	0.125	0.375 SYN
	EO (% *v*/*v*)	0.25	0.125	0.5		0.125	0.0625	0.5		1	0.25	0.25	
MDR-5 [non-APEC]	ENR (µg/mL)	8	0.25	0.03	0.53 ADD	8	1	0.125	0.625 ADD	8	1	0.125	0.375 SYN
	EO (% *v*/*v*)	0.25	0.125	0.5		0.125	0.0625	0.5		1	0.25	0.25	
MDR-6 [O18]	ENR (µg/mL)	4	0.25	0.06	0.31 SYN	4	0.25	0.06	0.31 SYN	4	0.125	0.03	0.155 SYN
	EO (% *v*/*v*)	0.25	0.0625	0.25		0.125	0.03	0.25		1	0.125	0.125	
MDR-7 [O78]	ENR (µg/mL)	4	0.5	0.125	0.375 SYN	4	1	0.25	0.5 SYN	4	0.5	0.125	0.25 SYN
	EO (% *v*/*v*)	0.25	0.0625	0.25		0.125	0.03	0.25		2	0.25	0.125	
MDR-8 [O1]	ENR (µg/mL)	4	0.0625	0.02	0.52 ADD	4	1	0.25	0.5 SYN	4	0.25	0.06	0.31 SYN
	EO (% *v*/*v*)	0.25	0.125	0.5		0.125	0.03	0.25		2	0.5	0.25	
MDR-9 [O78]	ENR (µg/mL)	2	0.5	0.25	0.5 SYN	2	0.5	0.25	0.5 SYN	2	0.25	0.125	0.375 SYN
	EO (% *v*/*v*)	0.25	0.0625	0.25		0.125	0.03	0.25		1	0.25	0.25	
MDR-10 [O78]	ENR (µg/mL)	2	0.5	0.25	0.5 SYN	2	0.5	0.25	0.5 SYN	2	0.25	0.125	0.375 SYN
	EO (% *v*/*v*)	0.25	0.0625	0.25		0.125	0.03	0.25		1	0.25	0.25	
SDR-1 [O78]	ENR (µg/mL)	1	0.25	0.25	0.5 SYN	1	0.5	0.5	0.625 ADD	1	0.25	0.25	0.5 SYN
	EO (% *v*/*v*)	0.25	0.0625	0.25		0.125	0.0156	0.125		0.5	0.125	0.25	
SDR-2 [non-APEC]	ENR (µg/mL)	1	0.25	0.25	0.5 SYN	1	0.25	0.25	0.5 SYN	1	0.03	0.03	0.28 SYN
	EO (% *v*/*v*)	0.25	0.0625	0.25		0.125	0.03	0.25		0.5	0.125	0.25	
SDR-3 [O78]	ENR (µg/mL)	0.5	0.125	0.25	0.5 SYN	0.5	0.125	0.25	0.5 SYN	0.5	0.06	0.125	0.375 SYN
	EO (% *v*/*v*)	0.25	0.0625	0.25		0.125	0.03	0.25		0.5	0.125	0.25	
SENS-1 [non-APEC]	ENR (µg/mL)	0.125	0.03	0.25	0.5 SYN	0.125	0.03	0.25	0.5 SYN	0.125	0.03	0.25	0.375 SYN
	EO (% *v*/*v*)	0.25	0.0625	0.25		0.125	0.03	0.25		0.5	0.06	0.125	
SENS-2 [O78]	ENR (µg/mL)	0.032	0.008	0.25	0.5 SYN	0.032	0.016	0.5	0.75 ADD	0.032	0.008	0.25	0.5 SYN
	EO (% *v*/*v*)	0.25	0.0625	0.25		0.125	0.03	0.25		0.5	0.125	0.25	
SENS-3 [O2]	ENR (µg/mL)	0.016	0.008	0.5	0.75 ADD	0.016	0.008	0.5	0.75 ADD	0.016	0.008	0.5	0.625 ADD
	EO (% *v*/*v*)	0.25	0.0625	0.25		0.125	0.03	0.25		0.5	0.06	0.125	
ATCC 25922 [non-APEC]	ENR (µg/mL)	0.016	0.008	0.5	0.75 ADD	0.016	0.008	0.5	0.75 ADD	0.016	0.008	0.5	0.625 ADD
	EO (% *v*/*v*)	0.25	0.0625	0.25		0.125	0.03	0.25		0.5	0.06	0.125	

MICi—individual minimum inhibitory concentration; MICc—MIC in combination: minimum inhibitory concentration of enrofloxacin in the presence of essential oil or minimum inhibitory concentration of essential oil in the presence of enrofloxacin; FIC—fractional inhibitory concentration; FICi—FIC index; non-APEC—*E. coli* serotype other than O1, O2, O18 or O78; Type of interaction: green—synergy (SYN), yellow—additive effect (ADD).

## Data Availability

The data presented in this study are available upon request from the corresponding author.

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
