# Peer review of "Interaction between Enrofloxacin and Three Essential Oils (Cinnamon Bark, Clove Bud and Lavender Flower)—A Study on Multidrug-Resistant Escherichia coli Strains Isolated from 1-Day-Old Broiler Chickens"

_ijms, 2024, doi:10.3390/ijms25105220_

Round 1

Reviewer 1 Report

Comments and Suggestions for Authors

The manuscript describes the combination of cinnamon bark, clove bud and lavender flower essential oils with enrofloxacin to evaluate the bacteriostatic activity against 16 strains of APEC. Although the authors analyzed by MIC and checkerboards and found that essential oils are synergistic with enrofloxacin, there are still some data that are not enough to justify this combination a good antibiotic alternative. After careful review, some comments are provided below:

1The authors evaluated the antibacterial activity of cinnamon bark, clove bud and lavender flower essential oils against APEC only from MIC, and the antimicrobial activity should also be evaluated from several aspects such as bacterial growth curve and time-kill curve analysis.

2The authors note that the combination lavender flower essential oil and enrofloxacin has a synergistic effect, but there is insufficient evidence that this effect is realized in animal models. Does long-term use induce resistance.

3What is the mechanism of synergy between lavender flower essential oil and enrofloxacin? Such as membrane permeability assaysproton motive force assaysefflux pump assaysin vitro protein synthesisin vitro transcription and mRNA translation.

4How safe is lavender flower essential oil

5Whether the taste and price of essential oils affect the use in production.

Comments on the Quality of English Language

Moderate editing of English language required.

Reviewer 2 Report

Comments and Suggestions for Authors

The manuscript describes the in vitro synergistic effect of enrofloxacin and essential oil combination against chicken E. coli. The manuscript was generally well-written. The objective was clear and the study design was appropriate. I only have minor comments to the authors.

1.      Title: The authors may consider removing the phrase “as a potential risk to human as consumer of poultry products” because this study did not investigate any risk to human consumers.

2.      Abstract: In addition to the MICs of essential oils, the authors please add the MIC data of enrofloxacin in the abstract as well.

3.      Line 67-70: In the Introduction, please provide a brief review of previous studies on combinations of antibiotics (especially enrofloxacin) and essential oil against pathogenic bacteria, and identify the research gap.

4.      Line 77-78: Because there is no result of the disk-diffusion experiment in Table 1, this sentence should be rewritten or deleted.

5.      Line 118-119: It is unclear what is the “typical enrofloxacin resistance”? Do the authors mean a minor or moderate degree of drug resistance? Please clarify.

6.      Line 131-158: This paragraph is not the results of this study. It should be moved to the Materials and Methods section.

7.      Line 362-377: This paragraph just provides general information on enrofloxacin and no mention or discussion of the results of the experiment at all. Therefore, it should be moved to the Introduction part rather than the Discussion.

8.      Discussion: The authors should discuss or propose an idea on the mechanism responsible for the observed synergism between enrofloxacin and essential oil. Also, the potential application in the fields should be emphasized.

9.      Conclusion: This part is a little bit long. The authors may remove unnecessary sentences like “Multidrug-resistant strains are becoming an increasing challenge” or “Due to some limitations.....their efficacy in general” or “and no neutral results or, what would be the worst, antagonism”.

Reviewer 3 Report

Comments and Suggestions for Authors

Manuscript ID: ijms-2942063

The manuscript authorised by Zych et al. raises an important issue regarding the search for alternative therapies based on essential oils to combat avian infections mainly caused by multidrug resistant strains (MDR) of Escherichia coli. After a careful perusal, however, I have detected important problems that should be properly addressed, including more evidence from enrofloxacin and EOs individual effects as well as their further interactions. In general, the conclusion is not well supported by data coming from MICs determinations. There are other flaws of conductance and presentation. In my opinion this article is too preliminary, and it cannot be recommended for publication. These are some important points that must be analyzed in detail.

-A precise description of the MICs procedures followed here is mandatory. My group has experience in MICs calculation and it is difficult to understand how single MICs may be determined in formulations of two or several compounds. Some results obtained from MDR strains show synergistic effects when combined with EOs, which are difficult to reconcile by themselves.

-As an example, MDR-1 (078) displays MICi of 64 (ug/ml) for ENR and of 0.25 (v/v) for cinnamon. Notably, however, their mixture induces about a mutual 4-fold decrease in the MICc two compounds, given raise to synergy. Similar data were obtained with the other two EOs. Surprisingly, these actions seem to be weaker with the E. coli sensitive strains. Because these natural products act through distinct mechanisms, a convincing explanation must be provided by the authors.

-Although the FICI (better the term “FIC index”) I s a useful mathematical tool for checking synergism, many interactions are just in the limit of synergism (< 0.5). Other approaches should be followed to confirm this point.

-The largest part of the text is devoted to explaining the data presented in the unique Table 1. It is important to report more information on the features of the distinct E. coli strains that belonging to the same phenotype display rather distinct MICs. Furthermore, to strength the hypothesis more evidence on viability effects (i.e. CFU counting) are required.

-The chemical composition of EOs varies among batches depending on abiotic factors and standardization is not always an easy task. Therefore, when available, any information for the bioactive principles recorded in the EOs should be indicated.

-I am disinclined to believe that for antibiotics “the mode of action depends on their concentration… (L. 166) or that EOs seem to have no specific cellular targets (l. 174)”. In fact, the Discussion is also too large, descriptive and it is somehow out of focus. The authors should revise these matters with detail.

Comments on the Quality of English Language

-The English style and grammar have to be carefully revised (i.e.: “was classifies…, Where Eos are concerned?).

Round 2

Reviewer 3 Report

Comments and Suggestions for Authors

This manuscript is a revised version of a previously submitted paper. I am the Reviewer 2 in charge of that peer-review. In their reply comments, the authors have addressed some questions suggested in my previous report. This effort is certainly appreciated. Unfortunately, I am still reluctant about the following essential point: the conclusions advanced in this study are not funded on clear and convincing experimental support.  A few queries:

-After a careful reading, I have serious difficulties to understand the checkerboard procedure, and how single MICs are calculated in complex mixtures of compound. The argument of reduction of concentrations is not valid, since conventional methods used lower doses to those tested here. Honestly, I truly doubt the interested readers can grasp this methodology.

-The concept of “weak synergism” must be clarified. The explanations provided are cursory, more relevant data are necessary.

-The quality of Figs. 1&2 is very poor and the information incomprehensible. Regarding the problems of scale, I wonder if the logarithmic plots might be helpful-

-The extension of Discussion is certainly inadmissible. Furthermore, the authors state that have decided to keep this section unchanged. Since this epigraph has been changed in several parts, I would appreciate to know exactly what they refer.

Comments on the Quality of English Language

It has been improved, but an additional survey is recomendable.

Round 3

Reviewer 3 Report

Comments and Suggestions for Authors

Please, see the enclosed comments for the Editors.

Comments on the Quality of English Language

The English style and presentation is generally correct with minor errors.